Dynamic trajectory index method based on large-scale real-time trajectory data

Zhai Huawei 1
Cui Licheng 2 clclicheng@163.com
http://orcid.org/0000-0003-1840-9958 Polat Kemal 3
Alenezi Fayadh 4
1 Information Science and Technology College, Dalian Martime University , Dalian, Liaoning , China
2 Public Security Information Department, Liaoning Police Academy , Dalian, Liaoning , China
3 Department of Electrical and Electronics Engineering, Bolu Abant Izzet Baysal University , Bolu , Turkey
4 Department of Electrical Engineering, Jouf University , Jouf , Saudi Arabia
Bhattacharyya Siddhartha
Electronic publication date: 2025 Apr 7
Publication date: 2025
Volume: 11
Electronic Location ID: e2785
Received 2024 Dec 16; Accepted 2025 Mar 5
Copyright: © 2025 Zhai et al.
Copyright year: 2025
Copyright holder: Zhai et al.
License: This is an open access article distributed under the terms of the Creative Commons Attribution License, which permits unrestricted use, distribution, reproduction and adaptation in any medium and for any purpose provided that it is properly attributed. For attribution, the original author(s), title, publication source (PeerJ Computer Science) and either DOI or URL of the article must be cited.
License URL: https://creativecommons.org/licenses/by/4.0/

Keywords: Dynamic trajectory index, Primary index, Core index

Funding: Liaoning Provincial Natural Science Foundation of China 2022-MS-420 This work was supported by Liaoning Provincial Natural Science Foundation of China (Grant No. 2022-MS-420). The funders had no role in study design, data collection and analysis, decision to publish, or preparation of the manuscript.

==============================
Constructing a trajectory index can efficiently improve the performances of trajectory data processing, provide basic supports for trajectory data mining. With the constantly growing of trajectory data scale and increasing demands for trajectory retrieval efficiency and accuracy, the indexing methods have become more and more crucial. The indexing method faces significant challenges in terms of spatiotemporal trajectory locality, imbalanced trajectory distribution and low trajectory data value density. To address these, we proposed an indexing method based on large-scale real-time trajectory data, it extends the vertical storage mode of HBase, designs the core index, and optimizes the design of the row key, refines the data retrieval process and provides specific mappings for each independent part of the dataset. Besides, it designs the primary index, implements a dynamic indexing mechanism, dynamically load relevant index based on query strategies to flexibly meet the complex query requirements. Comparative experiments demonstrate that the proposed index method is superior in range retrievals and trajectory retrievals, the responding speed is faster.

Introduction

With the wide application of sensor technology and mobile devices, large amounts of trajectory data are generated and captured every minute. These data record very valuable information, such as status information, location information and so on, they are very important for optimizing urban transportation, improving public safety and more. However, with the growing of data amount and increasing requirements, the existing data processing technologies have shown their limitations when facing the large-scale real-time trajectory data.

Motivated by the idea of analyzing multiple dimensions of trajectory data, many studies focus on the large-scale trajectory data indexing (Tan, Luo & Ni, 2012; Alarabi, Mokbel & Musleh, 2018; Tian et al., 2022), however, these works are mainly based on spatial (Xie et al., 2016; Ray, 2014) and spatio-temporal (Alarabi, Mokbel & Musleh, 2018; Eldawy, Alarabi & Mokbel, 2015) characteristics of trajectory data. The indexing models can deal with trajectory data, however, trajectory data suffers from imbalanced data distribution and low data value density, this results in these methods being far less efficient, and fails to meet real requirements. Besides, in processing real-time trajectory data, efficiently managing and updating data indexing is very important and difficult. Due to the increasing data, the common indexing methods are inefficient in frequently inserting data, it leads to increase the complexity of data management and reduce the speed of query response.

In summary, the article deeply researches these problems existing in the storage and retrieval of trajectory data, such as spatial and temporal anisotropy, frequently inserting data, inefficient index constructing and so on. A novel dynamic indexing method is proposed. It designs the range index based on geocoding and time information, and constructs the trajectory index with identifier extended TB-Tree, these can effectively improve the query efficiency for large-scale real-time trajectory data and meet complex queries. Besides, aiming at the shortcomings of existing methods in frequently inserting real-time data, we predefined a template indexing schema for both the range and trajectory indexes, it reduces the number of indexing rebuilds, improves the ability of handling real-time data. It also designs the core index, extends the vertical storage mode of HBase, and optimizes the row key, refines the data retrieval process. The structure of the core index is redefined, it introduces the partitioning identifier of pre-partitioning, which together with the moving object ID and time constitute the unique identifier of the trajectory, so as to solve the problem of indexing conflicts and imbalanced data distribution.

Related work

Existing indexing methods are mainly divided into three categories: one is indexes based on temporal dimension (including static index and dynamic index), second is indexes based on storage medium (including index based on HBase and index based on Cassandra), the third is indexes based on data characteristics (including multi-model index and learning model index).

Static index. Hadoop is typical static index constructing platform. The static indexes extending Hadoop are mainly used in historical spatial-temporal object index (STQuery (Li et al., 2016), ST-Hadoop (Alarabi & Mokbel, 2017), QaDR-tree (Han et al., 2016)), and historical moving object index (HadoopTrajectory (Bakli, Sakr & Soliman, 2019)), they meet high throughput requirements, but the response latency is high. In constructing static indexes, the common strategy is space first and time later; it segments the data space into multiple grids based on geographic locations, then the data within each grid is indexed by time, this effectively achieves the purpose of simultaneously indexing time and space. Wang (2023) designed the eXtend Z order (XZ) ordering spatial temporal index (XZST), it used XZ and eXtended Z-ordering for Time range (XZT) space filling curves to encode spatial sequence and time range of trajectories respectively, and then, the trajectory spatiotemporal encoding and related attribute information are concatenated in a certain order to form XZST indexes. Hughes et al. (2015) proposed a spatiotemporal data storage and retrieval engine, it used Z curve to encode the GeoHash, and effectively supported the retrieval of spatiotemporal trajectory big data.

Dynamic index. Dynamic index methods effectively resolve the problem of rapid growth of real-time trajectory data, meet immediate retrieval requirements. Distributed Index for high throughput Trajectory Insertion and Real-time temporal range query (DITIR) (Cai et al., 2017) is a distributed index, it is suitable for high throughput trajectory insertion and real-time temporal range query. DITIR assumes the data tuples arrive in the increasing order of their timestamps, and meanwhile, it introduces a template-based B+tree insertion schema to enable efficient indexing over fast data stream (Mazumdar et al., 2016). Distributed Trajectory R-(DTR)-Tree (Belhassena & HongZhi, 2017) is an R-tree index implemented on Apache Spark, it is widely used in trajectory and moving trajectory queries. Distributed Mining Trajectory R (DMTR)-Tree (Whitman et al., 2019) was proposed based on DTR-Tree, it can support the querying of moving trajectory by using DTR-Tree and inverted table. STARK (Hagedorn, Götze & Sattler, 2017) is built on top of Spark, it designs the grid partitioner and binary partitioner, and can efficiently perform join queries, K-NN queries and range queries for spatiotemporal data.

Based on HBase index. In recent years, many researches are carried out to use HBase to support multidimensional data querying, they are focus on designing the key ‘Row-Key’. UQE-Index (Huang & Chang, 2021) is an efficient index framework based on HBase, it constructs local R-Tree index, which supports efficient retrieval in region, so as to enable efficient inserting throughput and multidimensional querying. Nishimura et al. (2011) proposed a multidimensional data management system, MD-HBase, it bridges the gap between scale and functionality, and leverages a multi-dimensional index structure layered over a key-value. Zhang et al. (2018) proposed a hybrid index model for efficient spatio-temporal search in HBase, it takes both spatial and temporal information into consideration to effectively reduce the search space, and it is innovative in the field of spatiotemporal retrieval.

Based on Cassandra index. Cassandra database is an innovative distributed Not Only Structured Query Language (NoSQL) system, its indexing mechanism plays a key role in improving system performances. By storing mapping indexes between partition key and row in memory, it effectively supports fast access to SSTable, thus realizing rapid data location and reading (Zheng Xiong, 2020). Buyang & Feng Huasen (2021) proposed a Hilbert curve and Cassandra based indexing and storing approach for large-scale spatio-temporal datasets, it uses the S2 algorithm to reduce dimensionality of spatio-temporal data at storage, and designs to achieve efficient data indexing and querying by leveraging row key of Cassandra. Fu & Widagdo (2022) developed a system to perform spatial data retrieval using Structured Query Language (SQL) queries by using the PostGIS extension, it works by transforming an input SQL query into one or more Cassandra Query Language (CQL) queries to retrieve data from Cassandra, afterwards, the data is used to fill in the table declaration of the input SQL query.

Multi-model index. Multi-model index can more effectively handle complex retrieval, so, they receive more attention. JD Urban Spatio-Temporal Data Engine (JUST) (Li, 2020) is a spatio-temporal data engine developed for JingDong company, it uses HBase as the base storage layer, utilizes GeoMesa as indexing tool, and employs Spark as the execution engine. Meanwhile, it designs SQL engine and Software Development Kit (SDK) to extend its functions. TrajMesa (Li et al., 2020) is a trajectory storage system based on GeoMesa, it integrates various trajectory processing modules, and it can enable moving object queries and spatial range queries, and also optimizes the trajectory similarity queries. THBase (Qin, Ma & Niu, 2019) is a coprocessor-based scheme for big trajectory data management, it introduces a segment-based data model and a moving-object-based partition model to solve massive trajectory data storage, and exploits a hybrid local secondary index structure based on observer coprocessor to accelerate spatiotemporal queries.

Learned index. Learned index is the combination of artificial intelligence and database field, and attracts more and more attention. Ferragina, Lillo & Vinciguerra (2020) present the first mathematically-grounded answer to the open problem, that the learned indexes have significant advantages in theory over traditional indexing methods. Kraska et al. (2018) proposed the learned index framework and recursive model index. Galakatos et al. (2019) proposed a novel form of a learned index, FITing-Tree, it uses uses piece-wise linear functions with a bounded error specified at construction time, and allows the Database Administrator (DBA) to fit an index to a database and workload by being able to balance lookup performance and space consumption. Llaveshi et al. (2019) achieved data segmentation and accelerated the searching within nodes by combining the nodes of B+Tree with line regression model, and improved efficiency of index searches.

Real-time trajectory dynamic index

Real-time trajectory dynamic indexing method consists of two major parts: core index and primary index, the latter subdivided into the range index and trajectory index in further. The proposed index model is aiming to optimize the spatio-temporal feature retrieval of trajectory data, to accelerate data localization and improve query efficiency.

Core index

Generally, the vertical storage mode in HBase trajectory data storage strategies decomposes a complete trajectory into multiple records, so, using high table format for vertical storage, it is more conducive to the storage and query requirements of real-time trajectory data. For improving its performances in further, the core index table structure for dynamic indexing of real-time trajectory data was designed on the basis of vertical storage mode, as shown in the Fig. 1.

Figure 1 Core index table structure.

Where, row key (index) is the trajectory index of the core index table, its specific form is determined by the indexing strategy of the table. In terms of column design, in order to simplify data management, all trajectory data columns are stored in the same column family ‘cf1’, column ‘sid’, ‘time’, ‘(lon, lat)’, ‘version’ and ‘attrs’ respectively stores the information such as the trajectory ID, time, longitude and latitude coordinates, version no and other attributes.

Core index is the bridge connecting the primary index and data, its main function is to simultaneously meet different queries from the primary index, and also being able to differentiate and identify data. Based on the characteristics of trajectory data and corresponding partitioning method (Yu et al., 2024), the core index Hbase-based Trajectory Core index (HTC) for real-time trajectories is designed, its structure is shown in Eq. (1).

(1) HTC=shard+sid+t

where shard is the partitioning identification of pre-partitioning, it can solve the problems due to the spatial and temporal anisotropy of trajectory data, such as imbalanced data distribution after indexing, excessive data flow burden on a single HBase Region Server, and so on; sid is the ID of moving object, t is the time, they constitute the unique identifier of a trajectory together, which is used to solve the problem of indexing conflicts that may be caused by different trajectories under the same spatio-temporal encoding.

The workflow of core index is shown in Fig. 2. When it is to retrieval data, first, it accesses Zookeeper, then reads the information in the meta table of Zookeeper, and obtains the location of region by core index. Finally, after getting the corresponding region, it can locate the data exactly by the binary researching method. The data reading process does not start directly from the Hfile disk file deposited on Hadoop Distributed File System (HDFS), but first searches in memory, then tries to read data from the block cache, and finally retrieves it from the disk, thus ensuring the high efficiency of data retrieval. In Fig. 2, HTC is the core index, DN is the data node, RS is the partition server, Binary is the binary searching tree of each region, Hfile is the disk file.

Figure 2 Core indexing workflow.

Primary index

The key of primary index is to realize a dynamic indexing mechanism, it can dynamically load relevant indexes according to the query strategy, flexibly meets complex query requirements. So, for the spatial and temporal attributes of trajectory data, the range index combing with GeoHash and time and trajectory index with identifier extended TB-Tree are designed. The two indexes are built on the intrinsic encoding of HBase row keys, they maintain the integrity of original HBase, and also form a dynamic adaptive indexing mode, so as to retrieve the real-time trajectory data more efficiently.

Index template

For improving the efficiency of inserting real-time data, the template strategy (Dou et al., 2019) is introduced to construct primary index, so as to accelerate data inserting speed. The index template is a reuse mechanism that helps developers save a lot of repetitive operations.

The index template mainly includes five parts, its structure is shown in Table 1.

Table 1 The structure and configuration of the template index.

Configuration Name	Configuration	Note	
“order”	“1”	Template priority	
“template”	“tete”	Template matching name and method	
“settings”	“index”	Setting index	
“mappings”	“mytype”	Fields mapping in the index	
“aliases”	{…}	Alias of index	

Order. Sometimes, one template may not fully meet the requirements for creating a new index, it need to be fine tuned locally into a new index template, but repetitive operations can lead to a waste of time. So, template stacking and overlay can be used to avoid this waste, and this needs the template priority, the template priority is set by “order”, the larger its number is, the higher the priority will be.

Template. It shows the situation applied by the index template. For example, “template”: “tete*” shows that when constructing new indexes, all indexes start with “tete” will be automatically matched to the index template, and use it for corresponding settings and adding fields.

Setting. The “index” in the settings is to define the primary shard, replicas shard, refresh time, custom analyzer, etc. of the index. Its part structure is shown in Table 2.

Table 2 The structure and configuration of “index” in the template index.

Configuration Name	Configuration	Note	
“analysis”	{…}	Custom analyzer	
“numberofshards”	“m”	Number of primary shards	
“numberofreplicas”	“n”	Number of replicas shards	
“refreshinterval”	“T”	Refresh time	

The custom analyzer is key to the index template, and the custom index structure, such as range index and trajectory index, can be built by ‘analysis’.

Mappings. Field mappings include dynamic mappings and static mappings, its commonly used structure is shown in Table 3.

Table 3 The structure and configuration of “mytype” in the template index.

Configuration Name	Configuration	Note	
“dynamictemplates”	[…]	Dynamic mapping	
“properties”	{…}	Mapping response	

Range index

GeoHash (Ulu, Kilic & Türkan, 2024) is an encoding method for spatial data based on latitude and longitude, it converts geolocation information into a sequence of characters, this simplifies the process of data storage and queries, and also naturally maintains the proximity between geographic locations. Based on GeoHash and time stamp, the template strategy is used to construct the range index for real-time data, the time of the range index is only kept in hours, minutes and seconds, which can largely reduce the comparing time of indexes and thus improve the efficiency of data retrieval. The encoding rule of constructing range index node is shown in Eq. (2).

(2) node=GeoHash(x,y)|t

where (x,y) is the spatial location information, t is the time information of data. In order to improve the indexing efficiency, the original 12-bit encoding of GeoHash is reduced to a nine-bit encoding (with an error of about 2 m) by the GeoUtil tool, and then time information is appended directly after this nine-bit encoding. In addition, due to the unequal weights of temporal and spatial attributes in GeoHash, and also considering that GeoHash uses leftmost matching rule, the method proposed in the article does not encode the temporal information during the encoding process, but use the splicing method to reduce the intrusion of temporal dimension into non-temporal optimized retrieval.

The range index model is shown in the Fig. 3. Taking a testing data as an example, data format is “unique identifier, latitude, longitude, other characters, time”, the specific data is “32.563127, −117.070351, 0, −81, 40773.807974537, 2011-08-18, 19:23:29”. First, the latitude and longitude data are transformed into base-32 one-dimensional encoding, “wx4g573np2sk”, then, the precision is kept within the range of 4 m by utilizing the precision function provided by GeohashUtil, after removing the end characters, the spatial unique encoding can be gotten, “wx4g573np”, the nine-bit spatial encoding is as the spatial part of the range index. And the time “15:36:29” is used as the temporal part, finally, the range index after combining is wx4g573np|15:36:29, which is inserted into the prefix tree.

Figure 3 Range index model.

Trajectory index

After analyzing above, the range index can be used to improve the efficiency of range retrieval, but, because of the trajectories with many overlapping areas, the efficiency of the trajectory retrieval is not much better, thus, constructing trajectory index is necessary.

TB-Tree (Wang et al., 2023) is an access method considering the specialty of trajectory data, it aims to handle relevant type queries. It can seek an access method to strictly preserve trajectories, so, in the index, each leaf node should only contain line segments belonging to the same trajectory, which is suitable for trajectory retrieval. However, there are many similar trajectories in the same partition, the TB-Tree need to be extended and optimized, thus, Identifier Extended TB-Tree is designed in the article, showing in the Fig. 4.

Figure 4 Identifier extended TB-Tree model.

Comparing with traditional TB-Tree, the main difference is that each internal node n in the tree contains the complete trajectory ID set (that is TIDs) of all trajectory segments in the subtree rooted at node n. For example, the child nodes of the internal node n2 contain three segments, their trajectory IDs are 1, 7 and 9, respectively, and the three IDs are included in the TID set of node n2. The number of segments contained in the identifier extended TB-Tree is usually greater than the number of trajectories passing through the subtree, thus, retaining the TID set can achieve filtering node trajectories without traversing its child nodes. To construct indexes for trajectory segments, a tuple “ (tid,cid)” needs to be allocated, where tid is the trajectory ID, cid is the corresponding core index.

Constructing method of identifier extended TB-Tree consists of four steps, node partitioning, constructing node, constructing tree, optimizing and adjusting. Node partitioning is to determine the time range and corresponding trajectory data contained by each node in the tree, it needs to ensure that the partitioning time periods are continuous, and each time period has a certain connection with adjacent time periods. Each time period corresponds a node, it stores all trajectory TIDs within the time period. When constructing the tree, first, it needs to choose the initial node as the root node, (the starting time of the entire data range is taken as the root node in the article). Secondly, the unidirectional connection method is used to connect adjacent time period nodes in chronological order. Finally, according to the definition of TB-Tree, it is to adjust the connecting result to form a tree structure, and build the linked list between leaf nodes, so as to ensure fast access between data. The specific steps are shown in Algorithm 1.

Algorithm 1 Constructing identifier extended TB-tree.

  Input: trajectory dataset tracks	
  Output: Identifier Extended TB-Tree rootNode	
 1 BuildTBTree (tracks)	
 2 {	
 3    tracks.SortByTimestamp()	
 4    rootNode=CreateNode(tracks)	
 5    BuildTree(rootNode)	
 6 CreateNode (data)	
 7 {	
 8    node=new Node(data)	
 9    Return node	
10 BuildTree (data)	
11 {	
12    node=new Node(data)	
13    Return node	

Real-time dynamic trajectory index model

For efficiently handling various trajectory data queries, the dynamic trajectory index model is designed based on the core index and primary index, showing in Fig. 5.

Figure 5 Dynamic index model.

For the complex trajectory spatio-temporal data retrievals, the core index and primary index work together to accomplish the complex tasks. The primary index implements a dynamic indexing mechanism, it can dynamically load relevant indexes based on query strategies, and handle spatio-temporal retrievals involving different time spans and spatial ranges, as well as queries of specific moving objects. Besides, it provides a high efficient index structure for each kind of queries, which reduces the searching scope and time when searching relevant data in the dataset. The primary index carefully designs the row key structure, and refines the process of data retrievals, and provides specific mappings for each independent part of the dataset, so as to accelerate the data retrieval speed and reduce the reliance on the full table scanning.

Based on the proposed index model, the dynamic trajectory querying process is showing in Fig. 6. After receiving the query request, first, the system analyzes the request to identify the specific type and conditions of each query. Secondly, based on the request features, it is to select the corresponding primary index, for example, the spatio-temporal range query will be associated with the range primary index. Finally, the system uses the primary index to search the corresponding core index, which contains pointers to the actual storage location of the data.

Figure 6 Dynamic index query process.

Experiments

Experiment setup

Datasets. The GPS data from one city public transportation system are used. After processing, each record is organized into basic equal lengths and retains four fields, vehicle ID, latitude, longitude and time. The data preprocessing is mainly including: (1) identifying and deleting abnormal data; (2) supplementing missing data by the historical data (Shang, 2023).

Experiment environments. The experiments used Java and Scala programming languages, (jdk 1.9 and Scala 2.12), and also utilize Spark2 series framework. Besides, a cluster with one master node and three slave nodes was built.

Experiment method. The combined index method (Zhou, 2022), time single index method and HBase native technology were selected for the comparing experiments. Besides, for the fairness, some settings were set. (1) multiple experiments for each query are conducted to eliminate the impacts of caching on queries, and the average of 5–95% of the queries are taken as the final results. (2) the same programing language and mode are selected to implement comparative experiments, so as to reduce the impacts of code executing order on query results.

Evaluation metrics. We focus on evaluating the following metric in our experiments. Time/Response time: the runtime to finish the queries.

Range index testing

The range query is the typical scenario in large-scale real-time trajectory data retrievals, it involves retrieval of different data amount within specific spatial area, as well as queries over different time span for the given data amount. For testing the performances of the proposed index in real applications, the comparing experiments were conducted, the results are shown in Figs. 7 and 8.

Figure 7 Range retrieval under different data amounts.

Figure 8 Range retrieval under different time range lengths.

The comparative analysis of query performances for specific time and spatial range under different data amounts is shown in Fig. 7, with the increasing of data amounts, the retrieval time of all index methods increases. From the figure, the HBase native retrieval technology is weak in handling large-scale trajectory data, the larger the data amount, the longer the retrieval time, and the longest time is up to 80,000 ms, the main reason is that it relies on full table scanning, thus the retrieval latency increases significantly with the increasing of data amount. In contrast, the novel range index method optimizes the designing of row key, introduces the pre-partitioning identifiers, it reduces significantly the data area to be scanned, thus, it performs better than the time single index method and the combined method, although the retrieval time still increases with the increasing of data amount, the longest time does not exceed 14,000 ms, much lower than other indexing methods.

Under different time range, the range query experiments were conducted using different index strategies, the comparative analysis is shown in Fig. 8. From the figure, it revealed once again that HBase native technology spent more time in retrieving data due to its reliance on full table scanning, and it is especially in queries over the larger time ranges. In contrast, the single index method is based on time and reduces some searching volume, however, due to data spanning multiple partitions, retrieval performance is still limited. The combined method used the combined index of GeoHash and time, and achieves relatively higher retrieval efficiency through a more refined data filtering mechanism. The proposed index method utilizes the pre-partitioning of trajectories, similar trajectory storage strategy within the same partition and optimizes the design of the public row key, to efficiently reduce unnecessary partition retrieval and improve the retrieval efficiencies. Besides, the primary index is preserved in memory, it accelerates the responding time of data retrieval, so as to result in relatively better performances in all tested time ranges than others.

Trajectory index testing

The trajectory query plays an important role in the applications of large-scale real-time trajectory retrieval, it mainly includes two scenarios, one is to retrieve specific trajectories over a given time range under different data amounts, the other is to retrieve the single trajectory over different time ranges under fixed data amounts. Thus, the experiments were conducted to evaluate the performances of the proposed index model under these scenarios, the results are shown in Figs. 9 and 10.

Figure 9 Trajectory retrieval under different data amounts.

Figure 10 Trajectory retrieval under different time range lengths.

Figure 9 shows the performances of trajectory retrievals under different data amounts, the retrieval time of the four index models increases with the increasing of data amounts. HBase native retrieval mechanism lacks the direct support for trajectory concepts, its performance is not better, the retrieval time is longer, and more variable as the data amount increases, and the longest time is up to 64,000 ms. The time single index can reduce the data retrievals that are not related to the query time, and improve the retrieval speed, but the improvement is limited, the retrieval time is still very long. The combined index needs to conduct the secondary query to get the full data records, so, its retrieval speed is limited. In contrast, due to the features of the identifier extended TB-Tree, the proposed index can quickly locate the core index, effectively reduce data retrievals, and ensure high performances and low latency for the large-scale real-time trajectory data retrievals, its retrieval time is shortest, and does not increase significantly with the increasing of data amounts, the longest is only up to 12,000 ms.

From Fig. 10, because of the full table scanning, the retrieval time of HBase native technologies remains at a relatively high level. The time single index shows growing trend in retrieval time, because the data partitions will increase as the time range expands. The combined index can encounter the same situation as the time single index, but its performance is less affected by the time range. The proposed indexing method introduces the GeoHash coding and time to construct the range index, due to the index template strategy, its time part is only kept to seconds, meanwhile, it uses the GeoUtil tool to reduce the length of GeoHash coding to 9 bit, and does not encode the time part, reduces the time of comparing index, and it can achieve fast and accurate trajectory localization in each time range by the efficient management of the core index and primary index, so, it is better than other technologies.

Conclusion

The article deeply researched the real-time dynamic trajectories, considering the unique properties of trajectory data, as well as the data storing characteristics and data query requirements in HBase, the novel real-time dynamic index method is proposed, the index structure includes two parts, core index and primary index, and it introduces the template index strategy to optimize the insertion of real-time data. Comparative experiments show that the proposed index model is efficient and also superior in handling real-time trajectory data. The proposed indexing method is applied in the storage and retrieval of the urban public transportation trajectory data, it can improve the speed of range and trajectory retrieval, and provide basic data support for the public transportation scheduling, dispatching and other resource optimization, quickly and accurately complete the pre-preparation and temporary scheduling tasks. Although the inserting and retrieval of dynamic data is relatively better, it has not yet full considered the data deleting and updating mechanisms, which need to construct a more comprehensive dynamic indexing system.

Supplemental Information

Supplemental Information 1 Code including index, partition and retrieval.

Supplemental Information 2 Trajectory data.

Additional Information and Declarations

Competing Interests

The authors declare that they have no competing interests.

Author Contributions

Huawei Zhai analyzed the data, prepared figures and/or tables, authored or reviewed drafts of the article, and approved the final draft.

Licheng Cui conceived and designed the experiments, performed the experiments, performed the computation work, authored or reviewed drafts of the article, and approved the final draft.

Kemal Polat conceived and designed the experiments, prepared figures and/or tables, and approved the final draft.

Fayadh Alenezi analyzed the data, authored or reviewed drafts of the article, and approved the final draft.

Data Availability

The following information was supplied regarding data availability:

The data is available at Zenodo: Cui. (2024). TD20241210 [Data set]. Zenodo. https://doi.org/10.5281/zenodo.14349038.

The code is available at Zenodo: Cui. (2024). Computer code-Java code. Zenodo. https://doi.org/10.5281/zenodo.14493115.

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
