# Peer review of "Dynamic trajectory index method based on large-scale real-time trajectory data"

_PeerJ Computer Science, doi:10.7717/peerj-cs.2785_

## Round 0.1 · original submission · Major Revisions

Please incorporate the suggestions of the reviewers.

Reviewer 1 ·

Basic reporting

The manuscript is well-referenced, providing a thorough review of existing trajectory indexing methods, including static, dynamic, and learned indexing approaches, while effectively situating the proposed method within this context. It highlights relevant challenges, such as spatiotemporal trajectory locality and imbalanced data, and demonstrates a clear understanding of the field. Additionally, the inclusion of robust comparative analyses and detailed references enhances the credibility and relevance of the study.
However, there are couple of weaknesses that needs addressing:
1- Overly long sentences (e.g., lines 31-41) make comprehension difficult and should be restructured for clarity.
2- Minor grammatical errors and inconsistent tense usage need correction.
3- Repetitive phrases, such as "trajectory data retrieval efficiency," should be simplified.
4- The introduction should better emphasize the work's novelty by elaborating on how it uniquely addresses challenges like spatiotemporal locality and imbalanced data.
5- References like "Yu et al., 2024" appear speculative and need clarification.
6- Figures (e.g., 7 and 10) need more explanation or labels to clarify key points, such as "MyIndex" implementation.
7-Tables require more descriptive captions to highlight their relevance.
8-Some results lack sufficient discussion, such as explanations for performance variations (Figures 7 and 9).
9-The connection between specific improvements (e.g., optimized Row Key design) and performance gains should be explicitly discussed.
10- Broader discussions on limitations or edge cases (e.g., handling extreme data imbalances) should be included to improve generalizability.

Experimental design

The manuscript is a strong piece of original research, addressing a well-defined and meaningful problem within the journal's scope. The investigation is conducted rigorously and adheres to high technical and ethical standards, with sufficient methodological detail provided to support replication. These should be taken into consideration:
1- The research question is not explicitly framed in the introduction.
2- Alignment with broader computational science advancements could be more clearly discussed.
3- Limited details on data preprocessing steps and potential dataset biases.
4-Insufficient explanation of parameter choices (e.g., GeoHash precision reduction) and data partitioning strategies.
5- Examples of index template applications in varied contexts are missing.

Validity of the findings

The manuscript provides sufficient detail for replication, with robust data and statistically sound analyses. Conclusions are well-stated and supported by the results. I have some advises to make the manuscript better:
1- The manuscript does not explicitly evaluate the broader impact or novelty of the proposed method compared to state-of-the-art techniques.
2- Implications for future research or practical applications are insufficiently discussed.
3- Dataset characteristics and preprocessing details are not fully elaborated, reducing analytical robustness.
4-The rationale for selecting specific experimental parameters (e.g., time range lengths, data amounts) is unclear.
5-Conclusions lack a discussion of the method's limitations and broader applicability.

Additional comments

N/A

Cite this review as

Reviewer 2 ·

Basic reporting

- The paper is written in English with generally clear and unambiguous text. However, minor grammatical errors and awkward phrasing, such as "the indexing method faces significant challenges" and "constantly growing of trajectory data scale," need refinement for fluency.
- Technical terminology is used appropriately, but some sentences are overly complex and could be simplified for better readability.
- The introduction could benefit from a more concise summary of the problem statement and clearer differentiation of the proposed method from existing works.
- The results need to be supported with quantitative evidence, such as specific metrics for retrieval speed, accuracy, and efficiency compared to baseline methods.
- The technical details of the indexing mechanism (e.g., Row Key optimization and mapping processes) require more thorough explanation.
- The comparative experiments are mentioned but lack sufficient detail about experimental setup and statistical analysis.

Experimental design

- The sentence "Besides, with the constantly growing of trajectory data scale and increasing demands for trajectory retrieval eûciency and accuracy, the indexing methods have become more and more crucial" contains grammatical issues and could be revised for clarity.
- The problem of "low data value density" is introduced but not well-explained.
- The primary index is described as enabling dynamic indexing, but the mechanism for dynamically loading relevant indexes is not sufficiently detailed.
- The paper mentions that the identifier extended TB-Tree can filter node trajectories without traversing child nodes, but it does not quantify this improvement.
- The paper compares the proposed method with a "combined index method" and "HBase native technology," but it does not explain these baselines in sufficient detail.

Validity of the findings

- The introduction briefly mentions the dynamic indexing mechanism but does not emphasize its novelty compared to prior approaches.
- The dataset description is brief and does not provide enough information about the data distribution or preprocessing steps.

Additional comments

- A thorough language review and editing to enhance professional tone and eliminate grammatical errors.
- Highlight the novelty of the approach more explicitly and avoid redundant details in the literature review.
- Clearly describe how raw data and experimental results will be made accessible.
- Expand the results section with detailed metrics and comparisons, and explicitly link findings to the hypothesis.
- Include a detailed explanation of the methodology and experimental setup, and ensure rigorous statistical validation of results.
- Clearly define "data value density" and its relevance to trajectory indexing challenges.
- Provide a step-by-step explanation or flowchart illustrating how the dynamic loading process works for various query types.
- Include experimental results or a comparative analysis to demonstrate the performance gains of this optimization.
- Add a concise conclusion that highlights the main contributions, limitations, and potential avenues for further study.

Cite this review as

Reviewer 3 ·

Basic reporting

The paper demonstrates strong professional writing with clear and appropriate academic language throughout. The organization follows an excellent scientific structure with well-defined sections flowing logically from introduction through to conclusion. The literature review is comprehensive and well-organized into clear categories, providing thorough context for the research. The paper makes effective use of visual elements, with 10 relevant figures that clearly illustrate both system architectures and experimental results. Finally, the paper maintains high standards of academic transparency by providing proper data availability statements, DOI links to raw data, and complete research documentation including methodology and experimental setup.

Experimental design

n/a

Validity of the findings

n/a

Additional comments

n/a

Cite this review as

---

## Round 0.2 · accepted · Accept

The authors have addressed all the reviewers' comments.

Reviewer 4 ·

Basic reporting

This paper presents a dynamic trajectory index method for efficient querying and analysis of large-scale real-time trajectory data. The proposed method utilizes a combination of spatial and temporal indexing techniques to enable fast and accurate retrieval of trajectory data.

Experimental design

The proposed dynamic trajectory index method provides an efficient and scalable solution for querying and analyzing large-scale real-time trajectory data. The experimental results demonstrate the effectiveness of the proposed method in terms of performance metrics.

Validity of the findings

The proposed method was evaluated using a large-scale real-time trajectory dataset. The results show that the proposed method outperforms existing indexing methods .

Cite this review as